# Deep-Risk: Deep Learning-Based Mortality Risk Predictive Models for COVID-19

**DOI:** 10.3390/diagnostics12081847

**Published:** 2022-07-30

**Authors:** Nada M. Elshennawy, Dina M. Ibrahim, Amany M. Sarhan, Mohamed Arafa

**Affiliations:** 1Department of Computers and Control Engineering, Faculty of Engineering, Tanta University, Tanta 31733, Egypt; dina.mahmoud@f-eng.tanta.edu.eg (D.M.I.); amany_sarhan@f-eng.tanta.edu.eg (A.M.S.); m.Arafa@f-eng.tanta.edu.eg (M.A.); 2Department of Information Technology, College of Computer, Qassim University, Buraydah 51452, Saudi Arabia

**Keywords:** COVID-19 detection, mortality and severity risk, deep learning, machine learning

## Abstract

The SARS-CoV-2 virus has proliferated around the world and caused panic to all people as it claimed many lives. Since COVID-19 is highly contagious and spreads quickly, an early diagnosis is essential. Identifying the COVID-19 patients’ mortality risk factors is essential for reducing this risk among infected individuals. For the timely examination of large datasets, new computing approaches must be created. Many machine learning (ML) techniques have been developed to predict the mortality risk factors and severity for COVID-19 patients. Contrary to expectations, deep learning approaches as well as ML algorithms have not been widely applied in predicting the mortality and severity from COVID-19. Furthermore, the accuracy achieved by ML algorithms is less than the anticipated values. In this work, three supervised deep learning predictive models are utilized to predict the mortality risk and severity for COVID-19 patients. The first one, which we refer to as CV-CNN, is built using a convolutional neural network (CNN); it is trained using a clinical dataset of 12,020 patients and is based on the 10-fold cross-validation (CV) approach for training and validation. The second predictive model, which we refer to as CV-LSTM + CNN, is developed by combining the long short-term memory (LSTM) approach with a CNN model. It is also trained using the clinical dataset based on the 10-fold CV approach for training and validation. The first two predictive models use the clinical dataset in its original CSV form. The last one, which we refer to as IMG-CNN, is a CNN model and is trained alternatively using the converted images of the clinical dataset, where each image corresponds to a data row from the original clinical dataset. The experimental results revealed that the IMG-CNN predictive model outperforms the other two with an average accuracy of 94.14%, a precision of 100%, a recall of 91.0%, a specificity of 100%, an F1-score of 95.3%, an AUC of 93.6%, and a loss of 0.22.

## 1. Introduction

Nowadays, most healthcare systems rely on big databases containing accurate medical records that have been collected from medical centers and comprise the health information of patients such as diagnoses, demographics, medications, vital signs, problem lists, and laboratory data. Developing applications based on artificial intelligence (AI) techniques in order to utilize these medical big data has attracted the attention of many researchers in the past few years. Recently, many AI-based medical applications have been introduced to provide intelligent healthcare systems that can effectively perform multiple vital functions such as early prediction and disease diagnosis, clinical decision support, patient support, prognosis assessment, and mortality risk prediction for severe illnesses. In fact, physicians have greatly benefited from the emergence of these applications in the medical field.

AI has permeated almost every aspect of our daily lives, raising its performance to an even better level. One of the most important approaches of AI is machine learning (ML) [1,2]. ML algorithms utilize computational methods to simulate intelligent human ways of understanding and learning in order to make decisions based on the available historical data. They are concerned with developing systems that can learn from examples without being explicitly programmed. A higher amount of data can guide the learning process and enhance the quality of the employed machine learning algorithm. Deep learning (DL) is considered to be an evolution of machine learning methods as it requires more data to achieve more efficiency, quality, and comprehensiveness for deeper cases. DL can detect the complicated relationship between the input data and identify the important features without human intervention, which helps in building a more powerful learning model [3]. Healthcare is one of the areas that has benefited from DL in many ways, such as in the classification and prediction of diseases. Based on DL approaches, more accurate healthcare tasks can be performed, such as the one introduced in this paper: the mortality risk prediction for SARS-CoV-2 virus positive patients.

The convolutional neural network (CNN) is one of the first deep learning models introduced in the DL field, and since then, it has been used in many different applications such as pattern recognition, classification, and prediction [4,5]. It is based on the use of several layers in order to reduce the high dimensions of the training data from which the attributes are extracted automatically.

Long short-term memory (LSTM) is a type of recurrent neural network (RNN) that depends on improving the performance of the model by solving the vanishing gradient problem since this contains a memory block that stores weights values [6,7,8]. Each memory block consists of three gates that determine the state and the output of the block. These gates consist of a forget gate, an input gate, and an output gate. LSTM has shown great performance and is impressive in many applications that capture sequential information compared to traditional methods.

Currently, predicting mortality risk is considered as an important topic of research that aims to identify the potential risk factors as well as predict the probability of mortality for patients with severe diseases [9,10,11,12,13,14,15,16]. Investigating the key risk factors of mortality is critical for early mortality risk prediction. Several medical studies, such as [17], have exploited the prognosis for hospitalized SARS-CoV-2 patients. They evaluated the ECG readings (as an indicator of heart disease) of hospitalized SARS-CoV-2 patients at the time of admission to the hospital and after 7 days of hospitalization; they subsequently concluded that ECG is useful in identifying patients with possible clinical risk. There was also a significant association between abnormal ECG and major adverse events in patients with COVID-19. Artificial intelligence, specifically ML, can greatly help in the prediction of such mortality risk problems given the appropriate dataset to work with. Diseases such as chronic kidney disease, brain stroke, heart attack as well as COVID-19 have been the focus of research works concerned with mortality risk prediction in the recent years and which developed several prediction models with high-accuracy prediction values. Since the appearance of COVID-19, numerous AI-based applications that use different types of ML techniques have been developed to predict the mortality risk factors and severity for COVID-19 patients.

Surprisingly, the effectiveness of DL applications in predicting the mortality and severity of COVID-19 is not well-documented. Many studies have addressed mortality risk prediction for COVID-19 patients [1,18,19,20,21,22,23,24,25]. The current studies on mortality risk prediction for COVID-19, which employ clinical datasets, mostly utilize ML methods but rarely make use of deep learning. Moreover, the majority of such studies use small clinical datasets. In the current research, we use a clinical dataset in two different forms—its original CSV form and its converted image form—in order to increase the performance of the DL predictive models. The main contributions of this study are the following:Developing three deep learning predictive models with different architectures to predict the mortality risk for COVID-19 patients.Using the clinical features of COVID-19 patients to predict their chances of survival.Converting the original clinical dataset into images that would be used by one of the proposed predictive models and subsequently analyzing its performance.Conducting a comparative analysis with some previous studies that used well-known ML methods on the same dataset.Comparing our proposed work to previous studies that employed the deep learning CNN model with various meta-heuristic methods and datasets.

The remaining part of this paper is organized as follows. Section 2 presents related research works and the background on predicting mortality risk and severity. The details of the materials and methods, the dataset used in the study, data pre-processing, and the proposed deep learning models are described in Section 3. Section 4 incorporates the results, including the experimental parameters and the performance metrics for our proposed system, with comparisons to current state-of-the-art systems of mortality risk prediction. Comparative analyses and a discussion are elaborated in Section 5. Finally, conclusions along with possible future work are presented in Section 6.

## 2. Background and Related Works on Predicting Mortality Risk and Severity

Certainly, with the declining situation of COVID-19 patients, the timely identification of patients at high risk is imperative and vital. Important decisions can be taken based on this early identification, such as establishing more responsive healthcare systems, immediate intervention in more effective ways to protect against the risk of death, or the use of intensive care, which promotes an improvement in the health condition of patients and avoids the risk of death or more severe complications [1]. Moreover, it reduces the burden on the health systems and enables the decision makers to allocate the limited health assistance resources during periods of high morbidity to critically ill patients.

On the other hand, doctors have been seeking the best possible medicine dosages that would result in a fast and effective cure for the patients. Studies have been presented to help anticipate the effects of certain dosages in reducing mortality risk, as in [26], which investigated the possible association between different dosages of LMWH enoxaparin administration and mortality in hospitalized COVID-19 patients.

In [1], an integrated predictive model using ML algorithms was designed and developed to identify health risks and predict the mortality risk for COVID-19 patients. Support Vector Machine (SVM), Artificial Neural Networks (ANN), Random Forest (RF), Decision Tree, Logistic Regression, and K-Nearest Neighborhood (KNN) are just a few of the traditional machine learning techniques that were employed to achieve this goal. The data of more than 2,670,000 COVID-19-infected patients from 146 countries around the world were used to train this model, which comprised 307382 labeled samples. However, this dataset had to be pre-processed using a set of algorithms in order to extract new features and process missing data values, remove redundant and useless data elements, and identify the most useful features. The predictive model achieved an accuracy of 89.98% for the prediction of the mortality rate, using a neural network with a 10-fold cross-validation model. Moreover, the authors identified the most important symptoms and features associated with the COVID-19 disease.

The authors of [19] sought the identification of the most meaningful mortality risk markers in a dataset containing epidemiological, demographic, clinical, laboratory, and mortality data, which were collected from the medical records of 485 patients residing in Wuhan, China. A multi-tree XGBoost algorithm was used to rank the features based on their significance in order to determine the important discriminative biomarkers causing patient death. Their results indicated that Lactic dehydrogenase (LDH), lymphocyte, and high-sensitivity C-reactive protein (hs-CRP) were the most important biomarkers. The decision tree model accordingly uses the three indicated biomarkers in order to predict the mortality of a patient, with a dead or alive classification accuracy of about 90%.

In [27], a prognostic prediction model that was built using the XGBoost algorithm was developed for the prediction of the mortality risk in COVID- 19 patients. A dataset of 375 patients, including 201 survivors, from the Tongji Hospital in Wuhan was used in this study. The prediction model achieved a survival prediction accuracy of more than 90% and was based on the main clinical features of LDH, lymphocyte, and hs-CRP.

The XGBoost algorithm in [28] was used to develop mortality prediction models for COVID-19 patients. A dataset of 296 patients, including 277 survivors, from the First People’s Hospital of Jiangxia District in Wuhan was used to train the models. The features of age, history of hypertension, and coronary heart disease were used to create the clinical model that attained an Area under the ROC Curve (AUC) value of 83%. The features of age, oxygen saturation (SpO2), hs-CRP, neutrophil and lymphocyte count, aspartate aminotransferase (AST), D-dimer, and glomerular filtration rate were used to construct the laboratory model. The laboratory model outperformed the clinical model, with an AUC value of 88%. The authors recommended that the clinical models with limited information should only be used for the preliminary examination of high-risk cases.

The authors of [29] presented a prediction model that used the XGBoost algorithm to predict COVID-19 mortality risk in the New York City healthcare system. Their dataset contained 8770 patients, including 7656 survivors; the patient data were collected from the Mount Sinai hospital. The prediction model was able to achieve an AUC value of 0.86, whereas the detected risk factors were higher age, male gender, higher heart rate, higher respiratory rate, higher body mass index (BMI), and chronic kidney disease (CKD).

In [30], a mortality risk tool based on the XGBoost algorithm was established to predict COVID-19 mortality. Their dataset contained 3927 COVID-19 patients whose data were collected from six diverse centers, including 33 hospitals in Europe and the United States. The performance of the prediction tool was evaluated through three validation groups: Seville patients, Hellenic COVID-19 study group patients, and Hartford hospital patients. According to these validation groups, the prediction tool achieved AUC values of 0.92, 0.87, and 0.81, respectively. The study concluded that the mortality main risk factors were older age, low oxygen saturation, high levels of CRP, blood creatinine, and blood urea nitrogen (BUN).

Similarly, in [31], a simple-tree XGBoost ML model was developed for death risk prediction in COVID-19 patients. A dataset of 1270 COVID-19 patients, 984 of whom were admitted to the Sino French New City Branch and 286 to the Optical Valley Branch of Wuhan Tongji hospital, was considered in the study. The least absolute shrinkage and selection operator (LASSO) regression method was employed to discover the most important clinical features that cause the mortality risk in COVID-19 patients. The achieved performance of the death risk prediction was above 90%, 85%, and 0.90, respectively, for precision, sensitivity, and the harmonic mean of precision (F1 scores) values. In this study, the authors identified the major features of critical death risk for COVID-19 patients as disease severity, age, and serum levels of hs-CRP, LDH, ferritin, and IL-10.

In [32], an ML model based on the SVM algorithm was introduced to predict the serum biomarkers in COVID-19 patients with the greatest risk. A dataset from the University of Texas Medical Branch containing information from 398 COVID-19 patients, of which 355 were survivors, was used in the study to predict death 24 hours before it occurred. The SVM prediction model reached a sensitivity of 91% and a specificity of 91% with an AUC value of 0.93 for death prediction in COVID-19 patients. The effective serum laboratory factors determined by the authors were blood urea nitrogen, c-reactive protein, serum albumin, serum calcium, and lactic acid.

In the study conducted by [33], the SVM algorithm was utilized as a predictive model for the severe symptoms of COVID-19 patients. Their dataset contained the data of 336 COVID-19 patients collected from Shanghai Public Health Clinical Center. The prediction model determined only four features from among 220 clinical and laboratory features that had a great effect on its performance; these features were age, CD3 ratio, GSH, and total protein. The model realized an AUC value of 97.57%.

In [34], the SVM algorithm was used to develop a severeness detection model in COVID-19 patients. A dataset of 137 COVID-19 clinical patients from the Tongji Hospital affiliated with Huazhong University of Science and Technology, including 75 who were severely ill, was employed in the study. The SVM prediction model achieved an overall accuracy of 81.48% using only 28 of the 32 clinical features. The study indicated the features of the urine test and the blood test as candidate severeness factors.

Likewise, in [35], a machine learning predictive model based on the SVM was developed to predict disease severity in patients with moderate COVID-19. A dataset of 172 moderate COVID-19 patients admitted to the Cancer Center of Wuhan Union Hospital was used in the study. The performance results of the prediction model in terms of mean accuracy, sensitivity, and specificity were 91.38%, 0.90, and 0.94, respectively. In this study, only 6 features out of 22 were selected to obtain the best performance from the prediction model. These features were interleukin-6 (IL-6), high-sensitivity cardiac troponin I, procalcitonin, hs-CRP, calcium level, and chest distress.

As in reference [20], a multi-tree XGBoost algorithm nomogram predictive model was utilized to predict the mortality risk for COVID-19 patients. Their dataset contained information on 375 COVID-19-positive patients collected from Tongji Hospital. The patients were classified into the following groups: 124 were at low risk, 75 at moderate risk, and 176 at high risk. The prediction model attained an AUC value of 0.961 for the derivation cohort and 0.991 for the validation cohort. The top-ranked features identified in this study were age, lactate dehydrogenase, neutrophils, lymphocytes, and hs-CRP.

In [21], the authors utilized various supervised ML techniques to predict the mortality risk for COVID-19 patients and to recognize the vital features that cause mortality. A dataset of 370 infected patients containing 1766 datapoints from Tongji Hospital in Wuhan was used in the study. The importance of the features was determined using the XGBoost classifier, and feature selection was performed using a neural network. The obtained features were then used to develop different prediction models with the machine learning algorithms: neural network, SVM, logistic regression, random forests, XGBoost, and decision trees. The results showed that the neural network prediction model was the best, with an F1 score of 0.969, an accuracy value of 96.53%, and an AUC value of 0.989. The powerful features that were extracted for mortality prediction in the study were age, neutrophils, lymphocytes, LDH, and hs-CRP.

In [22], the authors developed a multivariable mortality risk predictive model based on the XGBoost algorithm that determines the survival chances of COVID-19 patients at admission; the model was applied to periods of 7 and 28 days. A dataset of 1393 infected patients from the clinical and laboratory records at admission, taken from six Apollo hospitals in India, was used in the study. To achieve the best performance, only 23 features of more than 65 were selected for the prediction model to get an AUC value of 0.88, an accuracy score of 0.97, and a precision value of 0.91. For the validation cohort, the performance was 0.782, 0.93, and 0.77, respectively.

Hu et al. [36] developed an early mortality risk prediction in COVID-19 patients based on machine learning. The prediction model was built using a sample of 183 patients from the Sino-French New City Branch of Tongji Hospital in Wuhan (115 survivors and 68 non-survivors of COVID-19). An additional 64 patients from the Optical Valley Branch of Tongji Hospital in Wuhan were employed to externally validate the final predictive model (33 survivors and 31 non-survivors of COVID-19). Patients’ medical records were mined for demographic, clinical, and pre-admission laboratory data. A set of ten approaches were initially tried in the study, and only five were proven to be effective. These approaches were logistic regression (LR), partial least squares regression, elastic net model, random forest (RF), and bagged flexible discriminant analysis (FDA). All of them had the same performance, with an AUROC value of 88.1%, a sensitivity of 83.9%, and a specificity of 79.4% for the validation set. The simplicity and great the interpretability of the LR model were the reasons it was chosen as the best final model. The study determined the features of age, hs-CRP level, lymphocyte count, and D-dimer level to be the most important features.

Zhao et al. [37] used several statistical methods to build a risk-score model for mortality and the intensive care unit (ICU) admission system. The authors examined 641 COVID-19-positive patient data (including 195 patients who had been admitted to the intensive care unit, of which 82 are deceased) at Stony Brook University Hospital. Patient characteristics, such as symptoms, comorbidities, and demographics, were compared to those of non-critical COVID-19 patients in order to determine the most significant predictors. Using ML and LR on the test dataset, the researchers were able to predict death with an AUC value of 0.83, and ICU admission with an AUC value of 0.74.

The authors of [38] introduced a predictive model to anticipate possible COVID-19 acute symptom development. The study used a dataset of 125 COVID-19 patients from Guangzhou’s Eighth People’s Hospital (93 mild and 32 severe). The model achieved an AUC value of 94.4%, a sensitivity of 94.1%, and a specificity of 90.2%. While there were 17 distinct differences between the moderate and severe groups upon admission, only four were identified to be linked with progression to a severe condition. These were comorbidities, respiratory rate, CRP, and LDH.

Zhou et al. [39] constructed a model to predict COVID-19 infection severity. The study used a dataset of 377 patients (172 severe and 106 non-severe) from Wuhan’s Central Hospital. It achieved an AUC of 87.9%, a specificity of 73.77%, and a sensitivity of 88.6%. Age, CRP, and D-dimer were linked with severity in COVID-19 patients. The N/L*CRP*D-dimer was also revealed to be a significant predictor of disease severity, where N/L is the ratio of neutrophils to lymphocytes.

The authors of [40] proposed a model to estimate infection severity in COVID-19 patients. The study used a dataset of 127 patients (16 severe) from Ningbo’s Hwa Mei Hospital. The risk prediction model was built with the LR algorithm and achieved an AUC value of 90.0%. A significant rise was detected in the severe group for neutrophil percentage, neutrophil–lymphocyte ratio (NLR), fibrinogen, sialic acid (SA), CRP, interleukin-6 (IL-6), IL-10, interferon-γ (IFN-γ), partial pressure of oxygen (pO2), and partial pressure of carbon dioxide (pCO2). High levels of IL-6, CRP, and hypertension could be considered as important risk factors for evaluating COVID-19 severity. Furthermore, the results indicated the great importance of IL-6 in monitoring severe cases, especially COVID-19 severity cases.

Gong et al. [41] constructed a risk prediction nomogram to identify patients whose cases would progress to severe COVID-19. The study used a dataset of 372 hospitalized Chinese patients. The feature selection was performed using the LASSO regression technique. The machine learning algorithms: LR, DT, RF, and SVM, were used to build the prediction models. The study identified seven risk factors for severe COVID-19: older age, higher LDH, CRP, direct bilirubin (DBIL), red blood cell distribution width (RDW), BUN, and lower albumin (ALB). The predictive model achieved AUC values of 91.2% and 85.3% for the training and validation cohorts, respectively.

The authors of [42,43] used a dataset of 336 seriously ill Chinese patients (34 of whom died) to construct a predictive model for COVID-19 mortality. The predictive model was developed using the multivariable LR algorithm, and it was able to achieve an AUC value of 99.4%, a sensitivity of 100.0%, and a specificity of 97.2%. The study linked the features of low lymphocyte ratio, BUN, and D-dimer to the cases of COVID-19 patient deaths.

Luo et al. [44] used a dataset of 1018 COVID-19 patients and developed a model that could make an early prediction of the in-hospital mortality of COVID-19 patients. The predictive model used univariate and multivariable logistic regression to identify the main risk factors that caused in-hospital death among COVID-19 patients. The study showed that the model was able to achieve better performance with an AUC value of 90.7% by combining the IL-6 levels (>20 pg/mL) and the CD8+ T cell counts (<165 cells/μL).

Li et al. [45] built a mortality risk prediction model for COVID-19 patients based on the basic health situation of the patient and other parameters such as age and sex. The study used two different datasets: the GitHub dataset and the Wolfram dataset. The GitHub dataset, after pre-processing, had 28,958 cases (530 deaths), while the Wolfram dataset had 1448 records (123 deaths). An autoencoder based on a neural network was used to build the predictive model. Its performance was evaluated in comparison to other training algorithms: LR, RF, SVM, SVM one-class models, local outlier factor, and isolation forest. The GitHub dataset had a precision problem and only contained general information on the patient; therefore, the prediction accuracy yielded by this dataset was not sufficient. The study showed that the death of COVID-19 patients is related to whether they have a chronic condition or symptoms of gastrointestinal, renal, cardiac, or respiratory problems.

Terwangne et al. [46] built an EPI-SCORE predictive model with the aim of enhancing the accuracy of the existing COVID-19 mortality risk score according to the WHO model. The EPI-SCORE model was built using a Bayesian network analysis. The study used a dataset of 295 COVID-19 positive RT-PCR patients from Epicura Hospital Center in Belgium. The model identified acute renal damage, age, LDH, lymphocytes, and activated prothrombin time (aPTT) as the most vital features for risk. The ROC curve index obtained by the WHO model was 83.8%, and the EPI-SCORE model outperformed it with 91%. As concluded by the study, a few clinical and laboratory features could be added to improve the accuracy of the COVID-19 EPI-SCORE model.

In a machine learning predictive study, the authors of [23] employed an aggregated COVID-19 global dataset and performed a meta-analysis of the existing research in this area. The results of the meta-analysis showed that the most important risk factors causing COVID-19 severity are: cerebrovascular disease (CEVD), chronic obstructive pulmonary disease (COPD), cardiovascular disease (CVD), type 2 diabetes, malignancy, and hypertension. They subsequently applied several ML classification approaches on an aggregated group of data, which concluded that COPD, CVD, CKD, type 2 diabetes, malignancy, hypertension, and asthma were the most important features when classifying the deceased and survived patients of COVID-19. Furthermore, they concluded that age and gender were the most significant predictors of mortality from the point of view of symptom–comorbidity combinations. Finally, the authors proved that Pneumonia-Hypertension, Pneumonia-Diabetes, and Acute Respiratory Distress Syndrome (ARDS)-Hypertension were the most significant mortality risk factors of COVID-19 patients.

The authors of [24] conducted a study comparing the ability of some common ML algorithms to predict COVID-19 in-hospital mortality. The outperformed model was used to determine the in-hospital mortality risk factors in COVID-19 patients and also to design a predictive tool that could determine the in-hospital mortality. The study was conducted on COVID-19 patients diagnosed by PCR test at the COVID-19 referral center in the Veneto region in Italy. Thus, a dataset comprised of 341 patients was created; the median age was 74, and the most predominant gender was male. The ML algorithms compared in study were the recursive partition tree (RPART), SVM, RF, and the gradient boosting machine (GBM). The performance results were recorded according to sensitivity, specificity, and ROC curve metrics. The results showed that the RF algorithm had the best performance, with an ROC of 0.84 (95% C.I. 0.78–0.9). The study concluded that the most relevant risk factors of in-hospital death were age, vital signs (oxygen saturation and the quick SOFA), and lab parameters (creatinine, AST, lymphocytes, platelets, and hemoglobin).

In [25], a deep learning model built using CNN combined with an autoencoder (AE) was introduced to predict the survival probability of COVID-19 patients. The authors used a clinical dataset collected from publicly available resources. As the size of the collected data was relatively small, an AE was used to perform data augmentation and to generate a balanced dataset. The feature selection of the clinical dataset was performed using meta-heuristic algorithms: artificial bee colony (ABC), ant colony optimization (ACO), butterfly optimization algorithm (BOA), elephant herding optimization (EHO), genetic algorithm (GA), and particle swarm optimization (PSO).Their predictive model reached an accuracy of 96.05% as compared to 92.49% for the CNN model. The authors also trained their predictive model using another dataset to ensure the generality of the augmentation method. Moreover, the effect of the various clinical features on the mortality rate was studied, along with the correlations between the feature pairs. The commonly reviewed studies on the severity and the prediction of mortality risk in COVID-19 patients are summarized in Table 1.

As evidenced by the reviewed studies that used the clinical datasets for the mortality risk prediction of COVID-19, machine learning methods had been used more frequently than the deep learning ones. In addition, the majority of these studies used a small clinical dataset in its original form. To overcome these drawbacks, we decided to convert the clinical dataset records from their original form into a corresponding image form. The dataset created was presented in two ways. The first version is the original form, which contains 12,020 records of patients who tested positive for COVID-19. The second form was created by turning every record from the first form into images, with each image representing a row of data in the initial clinical dataset. The original data (Tabular data) were used as they were in the CNN model, and these data were converted into images before being used as inputs for the 2-dimensional CNN in the IMG-CNN model. The main reason for converting the clinical dataset into images was to be able to use the 2-dimensional convolutional neural networks rather than the 1-dimensional convolutional neural networks. The 2-dimensional convolutional neural networks have many advantages, which include extracting spatial features from the data and creating a robust network for classification. Since tabular data do not have a spatial relationship between their features, they were preferred for the conversion into images to create more stability in the CNN architecture. In our study, we developed a deep learning framework that used the clinical dataset in both its original and image forms in order to fill this gap.

## 3. Materials and Methods

We propose three deep learning-based mortality risk predictive models to determine the survival chances of COVID-19 patients by using their clinical features. The block diagram of the proposed predictive models is given in Figure 1. As illustrated in the block diagram, data collection is the first step in the procedure as we used a public clinical dataset. The collected dataset is then pre-processed, which creates two forms that represent the clinical dataset. The first form is the original form, which consists of 12,020 records of COVID-19-positive patients, and it is used by the first two proposed models. The second form is obtained by converting all the records from the original form into images, where each image represents a data row in the original clinical dataset. This image form is then used by the third proposed model.

The next step is the prediction process, where we use three predictive deep learning models to achieve the prediction task. The first model, denoted by CV-CNN, is a CNN model in which a 10-fold cross-validation technique is applied to the original clinical dataset, partitioning it into a training set and a test set. The second model, denoted by CV-LSTM + CNN, is a hybrid model based on the combination of LSTM and CNN. Similarly, as in the CV-CNN model, a 10-fold cross-validation is applied to the original clinical dataset, partitioning it into a training set and a test set. The third model, denoted by IMG-CNN, is a CNN model in which the converted images of the clinical dataset are used. The final step is the classification and performance evaluation of the three predictive models.

### 3.1. Datasets for the Study

In our study, a dataset of more than 12,020 COVID-19-positive patients is used. The data on positive male and female COVID-19 patients were collected from 146 countries around the world (from official government sources), with the average age being 44.75 [47]. The virus is verified through the detection of its nucleic acid. The original dataset consists of 32 data elements from each patient, which contain physiological and demographic data. To avoid biasing the results, the data samples from the training dataset are balanced for the number of both recovered and deceased patients [47]. A snapshot of the original clinical dataset is shown in Figure 2.

As in [1,46], we extract a total of 112 features from the original dataset: 80 features from symptoms and doctors’ medical notes concerning the health status of the patient, and 32 features from patient’s demographic and physiological information. From among these 112 features, only 57 are selected as the most useful and effective features. The selected 57 features are listed in Table 2, and they are sorted into symptoms, pre-existing conditions, and demographics [1].

### 3.2. Dataset Pre-Processing

The pre-processing stage is usually performed to prepare the collected data in order to satisfy the requirements of the deep learning models. In our work, the balanced dataset introduced in [1] was used as the CSV input data file for the CV-CNN model and the CV-LSTM + CNN model. The input dataset is recorded in the form tabular rows, as shown in Figure 3. Each row has 56 attributes, such as age, sex, hypoxia, etc. The number of rows represents the number of patients in the dataset, which contains 12,020 patients. Furthermore, a new representation of the input data is introduced by converting each row in the tabular input dataset into a 2-dimensional image that contains 9 × 6 pixels [48]. In transforming the tabular data into images, the numbers are converted to pixel colors ranging from black to white. The image appears with a black part and a white part based on the values of the numbers in the data. These converted images are then used as inputs for the IMG-CNN model. Samples of the converted images are shown in Figure 4.

### 3.3. The Proposed Deep Learning Predictive Models

In our work, three supervised deep learning predictive models are utilized to predict the mortality risk for COVID-19 patients. One of these predictive models is built using a combination of LSTM and CNN networks, while the other two models are constructed using only CNN. The clinical dataset in its original CSV form is used by the first two predictive algorithms. The last one uses a dataset that contains the converted images of the original clinical dataset. The ability of each one of these predictive models to predict the recovered and deceased cases of COVID-19 patients are investigated through the experimental results.

#### 3.3.1. The First Predictive Model: CV-CNN

The first proposed predictive model (CV-CNN) is a CNN model that is trained using the clinical dataset based on the k-fold cross-validation (K-fold CV) approach, with k = 10. The 10-fold CV randomly partitions the original dataset into 10 subsets of equal size. Of the 10 subsets, 9 are used as the training dataset to train the model, while a single subset is used as the validation dataset to test the model. The training subsets represent 90% of the whole dataset and are used to derive the model parameters. The validation subset represents 10% of the whole dataset and is used to evaluate the model. As we have 10 subsets, this means that the cross-validation process is repeated 10 times (folds). In each fold, a different subset is used for validation, as illustrated in Figure 5. The schematic diagram and the pseudo-code of the CV-CNN model are shown in Figure 5 and Figure 6, respectively.

The detailed configuration of the parameters and the architecture of the CV-CNN model is described in Table 3. The first three layers are convolutional layers, and they are used for feature extractions. These are followed by a maximum pooling layer. The output is then passed to a flatten layer, some dense layers, a batch normalization layer, and dropout layers. The overall number of parameters is 1,167,361, with 1,167,169 trainable parameters and 192 of non-trainable parameters.

#### 3.3.2. The Second Predictive Model: CV- LSTM + CNN

The second proposed predictive model (CV-LSTM + CNN) is developed by combining the LSTM with a CNN model. Similarly, as in CV-CNN, it is trained using the clinical dataset based on the 10-fold CV approach for training and validation. The original dataset is randomly partitioned into 10 subsets of equal size using the 10-fold CV. There are 10 folds to be executed, and a different validation subset is used in each fold, as illustrated in Figure 7. The schematic diagram and the pseudo-code for the CV- LSTM+CNN model are presented in Figure 7 and Figure 8, respectively.

Table 4 contains the full configuration of the parameters and the architecture of the CV-LSTM + CNN model. We use different types of layers, such as the batch normalization layers, reshape layer, time distribution layers, convolutional layers, flatten layers, dense layers, and drop out layers. The total number of parameters is 995,077, with 994,563 trainable parameters and 514 non-trainable parameters.

#### 3.3.3. The Third Predictive Model: IMG-CNN

The third proposed predictive model (IMG-CNN) is a CNN model, and it is trained using the converted images of the clinical dataset, where each image corresponds to a data row from the original clinical dataset. The dataset containing clinical images is randomly partitioned into 20% for training and 80% for validation. Figure 9 and Figure 10 illustrate the schematic diagram and the pseudo-code for the IMG-CNN model, respectively. The complete information on the parameters and the architecture of the IMG-CNN model is listed in Table 5. It starts with the 2-dimensional convolutional layers that are used for feature extractions, followed by a maximum pooling layer; the output is then passed to a flatten layer and then to some dense layers as well as batch normalization, activation, and dropout layers. The total number of parameters is 170,280,001, with 170,279,361 trainable parameters and 640 non-trainable parameters.

## 4. Results

### 4.1. Experimental Parameters

Python 3 and the Keras framework were utilized during the development of these models. They were tested by operating them on the Google Colab pro version [49], which has a P100 Graphical Processing Unit (GPU) processor, 2 terabytes (TB) of storage, and 25 gigabytes (GB) of random access memory (RAM). An optimizer and appropriate fit functions were utilized during the training and validation stages of the development of these models. In addition, each model went through about 1000 epochs, with a batch size of 32.

The findings were achieved by applying the performance metric equations to the outputs of the generated validation data, and the registered results indicate the highest validation values that could be attained. The Adam optimizer [50] was used for the training of the proposed model. A value of 0.0001 was assigned to the learning rate. The entire source code for the models that we generated can be seen on the GitHub website, at the location indicated by [51].

### 4.2. Performance Metrics

The proposed model’s performance was evaluated based on: accuracy, recall, precision, f1-score, and finally, Area Under Curve (AUC). Correspondingly, the confusion matrix was introduced for each model. The accuracy, given in Equation (Equation 1), is the number of instances where an accurate prediction was made based on the total number of instances [52].
(1)$$Accuracy=\frac{Tp+Tn}{Tp+Tn+Fp+Fn}$$
where Tp and Tn are parameters that represent the genuine positive and negative values, respectively. The false positive and false negative values are denoted by the notations Fp and Fn, respectively. Sensitivity or Recall is the number of actual samples that have been forecast as positive from the total number of samples that are in fact positive. It is also known as the true positive rate and is provided by Equation (Equation 2). In contrast, the true negative rate, which formerly went by the name of Specificity and is represented by Equation (Equation 3), is the number of samples that were genuinely negative and were correctly projected to be negative based on the total number of samples that were negative.
(2)$$Recall=\frac{Tp}{Tp+Fn}$$
(3)$$Specificity=\frac{Tn}{Tn+Fp}$$

Precision, which is sometimes referred to as the Positive Predictive Value [48], is denoted by Equation (Equation 4) and indicates the ratio of the number of samples that were truly positive to the total number of samples that were expected to be positive. The equation that displays the harmonic mean of precision, which is also referred to as the F1-score, is denoted by Equation (Equation 5).
(4)$$Precision=\frac{Tp}{Tp+Fp}$$
(5)$$F1-score=\frac{2*Tp}{2*Tp+Fp+Fn}$$

In addition, the most recent research lends support to the utilization of confusion matrix analysis in model validation [48] due to the fact that it is able to categorize data relationships and any distribution. It offers further information regarding the illustrative models of classification.

### 4.3. Deep-Risk: Deep Learning-Based Risk Mortality Prediction System Results

After introducing the three proposed deep learning predictive models for risk in COVID-19 patients, we now present our results by examining the performance of the three models in terms of precision, recall, F1-score, and accuracy using the dataset given in [1]. A complete analysis of these models is given, which indicates their performance in predicting recovered and deceased patients. The confusion matrix is also given for each model, and the behavior of the model during training through epochs are presented as graphs for precision, recall, loss, AUC, and accuracy. The results of the proposed Deep-Risk models are presented, followed by a brief discussion and an analysis of each proposed model.

#### 4.3.1. CV-CNN Model

Based on the experimental results, the confusion matrix for the CV-CNN model is shown in Figure 11. The figure clarifies that the model can successfully classify the two statuses, died and recovered, at a high ratio. In addition, the different evaluation metrics for each fold based on a 10-fold cross-validation of this model, which used precision, recall, f1-score, and accuracy between the training of the 10 folds, are introduced in Table 6. The performance metrics of precision, recall, and f1-score along with the recovered, died, macro average, and weight average values are represented in the table.

#### 4.3.2. CV-LSTM + CNN Model

Figure 12 displays the CV-LSTM + CNN model confusion matrix, which demonstrates that the model can classify the died and recovered cases at 94.23% and 76.43%, respectively. Additionally, the different evaluation metrics for each fold of a 10-fold cross-validation are shown in Table 7.

#### 4.3.3. IMG-CNN Model

In Figure 13, the confusion matrix for the IMG-CNN model illustrates the classification of died and recovered statuses, with the highest ratio being for the recovered cases’ images (100%) and the died cases’ images (83.66%). Table 8 shows the evaluation metrics for the IMG-CNN model. Similarly, Figure 14 shows the loss, precision, accuracy, AUC, and recall between the training and validation phases, with the number of epochs being equal to 1000 epochs.

As clearly shown by the model results and as illustrated in the models’ confusion matrix, the IMG-CNN model outperforms the other proposed models, the CV-CNN and the CV-LSTM + CNN models.

### 4.4. Proposed Models Comparison and Results Discussion

The current research focused on the use of deep learning systems to predict the survival chances of COVID-19 patients. We used patient data consisting of 12,020 records from the clinical dataset to construct and evaluate the proposed models [1]. First, this clinical dataset in its original numerical form was used to train and test two of the proposed models (CV-CNN and CV-LSTM + CNN). Then, all these records were converted into images such that each image represented one row of data from the original dataset; the images were subsequently used to train and test the third proposed model (IMG-CNN).

The experimental results of the three proposed Deep-Risk models are summarized in Table 9. The results of the CV-CNN and the CV-LSTM + CNN models for each performance metric were computed as the average of all the 10 folds. As clearly seen in Table 9, the IMG-CNN model is much better than the other two proposed models in terms of accuracy, precision, F1-score, and accuracy. The use of the image form in the clinical dataset produces better results than when the original dataset is used, especially with deep learning models.

After examining the results, we found that there were significant differences in performance between the CNNs trained on clinical data and those trained on image data. The reason for this is that the deep learning approach can identify the features presented in the input data more precisely when the input is an image as the operations performed on the input works better with images. This explains why converting the data into images and using it in the IMG-CNN model yields better results. Accordingly, the proposed IMG-CNN model will be used in the next section and compared with the previous models presented in previous work.

## 5. Comparative Analysis with State-Of the-Art Work

In our comparative analysis, the previous studies concerned with predicting the survival chances of COVID-19 patients were divided into two aspects: First, a comparison with previous works based on machine learning methods using the same dataset. Second, a comparison with previous works that used the CNN deep learning model but with different descriptive methods and different dataset. In addition, we also present an analysis of the experimental results to evaluate the prediction ability of our model.

### 5.1. Dataset-Based Comparison with State-Of the-Art Work

As discussed in the literature review section, very few methods have focused on predicting mortality with the use of clinical data. Additionally, existing methods have used features that are different from the ones employed in our experiments. In this section, we compare our work with a recent study that used ML approaches such as Support Vector Machine, Neural Network, KNN, LR, DT, and Random Forest on the same dataset [1]. As can be concluded from the results presented in Table 9 in the previous section, the proposed IMG-CNN model shows better performance than the other proposed models; therefore, we will use only the proposed IMG-CNN model in the comparison with the previous ML models that used the same clinical dataset [1].

Table 10 shows a comparison of the performance evaluation metrics of the different ML models presented in a previous study [1] (as reported in their paper) and our proposed IMG-CNN model using the same clinical dataset, as also represented in Figure 15. In the present paper, we evaluated the proposed model in terms of accuracy and AUC. The accuracy reached by our IMG-CNN model is 94.14% and the AUC is 93.6%; in comparison, the best results reported in [1] for the NN model were 89.98% and 93% for accuracy and AUC, respectively. The results show that our proposed IMG-CNN model outperforms all the existing models by a significant value, which demonstrates the effectiveness of our predictive model. This is due to the fact that deep learning-based models are more accurate than machine learning-based models as they are able to extract the important features of the input data in an accurate and automatic manner.

### 5.2. Deep Learning-Based Comparison with State-Of the-Art Work

There are few methods that have studied mortality rate prediction based on clinical data with the use of deep learning models. However, the features and the clinical dataset that were used in our experiments differ from those used in the introduced study [25].

Because the proposed IMG-CNN model outperforms the other two proposed models, as shown in Table 9, it will be used in the comparison of the previous studies that used deep learning models. Table 11 shows the performance evaluation metrics of the various deep learning models presented in a previous study [25] (as reported by their paper) and our proposed IMG-CNN model; the metrics presented are accuracy, precision, recall, specificity, F1-score, AUC, and loss. Figure 16 and Figure 17 show these results in a graphical form for a clearer visual comparison. The conducted experiment results reveal that our proposed IMG-CNN model outperforms the previous studies by a significant percentage in terms of accuracy, precision, specificity, and AUC. It achieved the highest accuracy (94.14%), precision (100%), specificity (100%), and AUC (93.6%), and the loss was minimal. This is due to the better representation of the features in the image form, which allowed the deep learning model to identify the correlation between the input features of the data that produced better prediction ability.

## 6. Conclusions and Future Work

In this research work, a supervised deep learning system for COVID-19 risk mortality prediction was developed; this system is called Deep-Risk. Three different deep learning models were designed for this purpose, namely the CV-CNN model, the CV-LSTM + CNN model, and the IMG-CNN model. Each model was trained and tested to see how well they would be able to identify the cases that had recovered and those that had died. The same clinical dataset was used for all three models; a single but 10-fold cross-validation was used with the first and second models, while a normal training method was used with the third one.

In developing our models, the CNN model and a combination of the CNN and Long-Short-Term Memory (LSTM) deep learning models were used. Instead of using the clinical dataset in CSV form, each record in the dataset was converted into an image and then used as input for the third model. Accuracy, precision, recall, specificity, F1-score, AUC, and loss were used as the performance metrics. It was discovered that the IMG-CNN model had a higher average accuracy (94.14 percent) than the CV-CNN model (85.27 percent) and the CV-LSTM + CNN model (CV-LSTM + CNN model) (86.06 percent). To demonstrate the generality of our Deep-Risk system, a comparative analysis with previous studies that used machine learning methods to analyze the same dataset as ours was presented. An accuracy score of 94.14% and an AUC of 93.6% show that our IMG-CNN model outperformed those of the previous studies. To further our investigation, we compared our proposal to previous work that used a deep learning CNN model with the use of different meta-heuristic methods and different datasets. In terms of accuracy, precision, recall, specificity, F1-score, AUC, and loss, our proposed IMG-CNN model outperformed those of the previous studies, with values of 94.14%, 100%, 91.0%, 100%, 95.3%, 93.6,% and 0.22%, respectively.

Based on the experiment conducted in this study, the following research directions are suggested:Apply different large datasets and deeply analyze the effectiveness of converting these datasets into images;Try to improve the performance of the predictive models by using other deep learning methods.

## Figures and Tables

**Figure 1 diagnostics-12-01847-f001:**
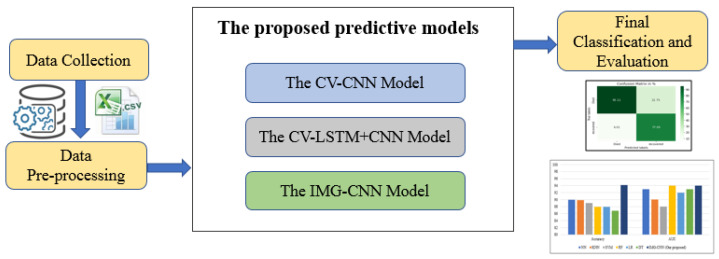
Block diagram of the proposed predictive models (Deep-Risk).

**Figure 2 diagnostics-12-01847-f002:**
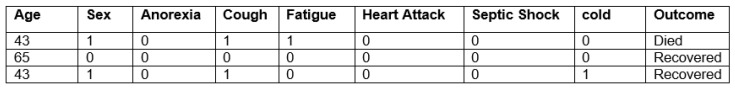
Sample of the tabular dataset.

**Figure 3 diagnostics-12-01847-f003:**
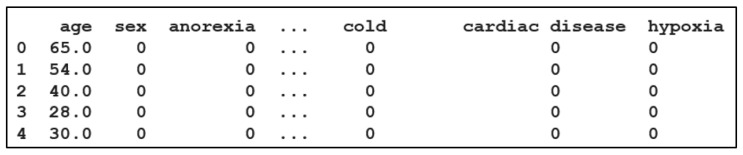
Sample of the original clinical dataset in tabular rows.

**Figure 4 diagnostics-12-01847-f004:**
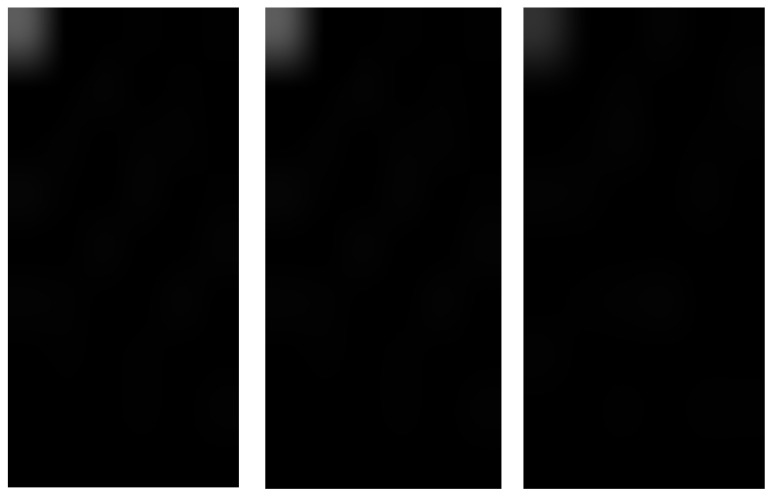
Samples of the converted images from the original clinical dataset.

**Figure 5 diagnostics-12-01847-f005:**
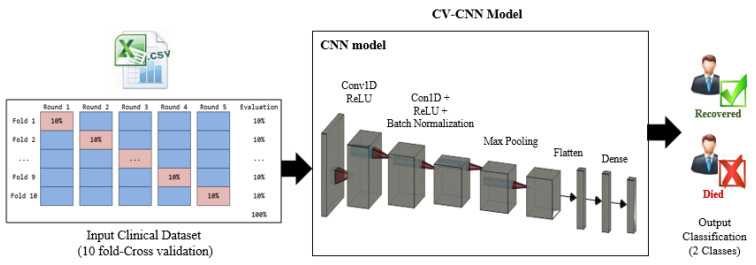
A schematic diagram of the CV-CNN model.

**Figure 6 diagnostics-12-01847-f006:**
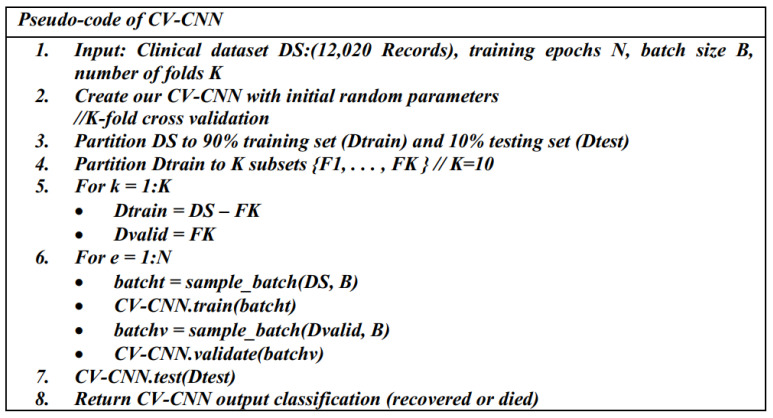
The pseudo-code of the CV-CNN model.

**Figure 7 diagnostics-12-01847-f007:**
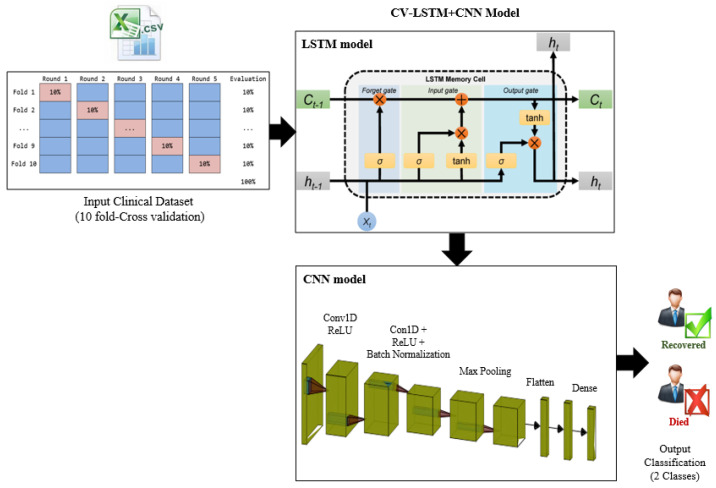
A schematic diagram of the CV-LSTM + CNN model.

**Figure 8 diagnostics-12-01847-f008:**
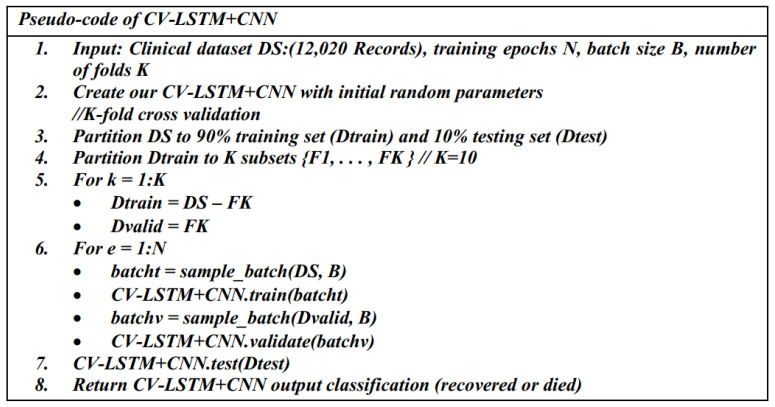
The pseudo-code of the CV-LSTM+CNN model.

**Figure 9 diagnostics-12-01847-f009:**
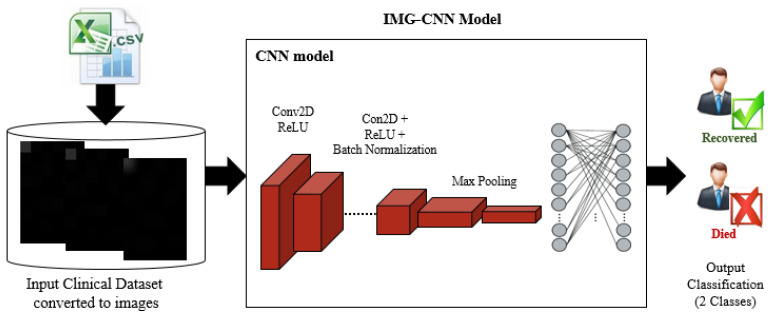
A schematic diagram of the proposed IMG-CNN model.

**Figure 10 diagnostics-12-01847-f010:**
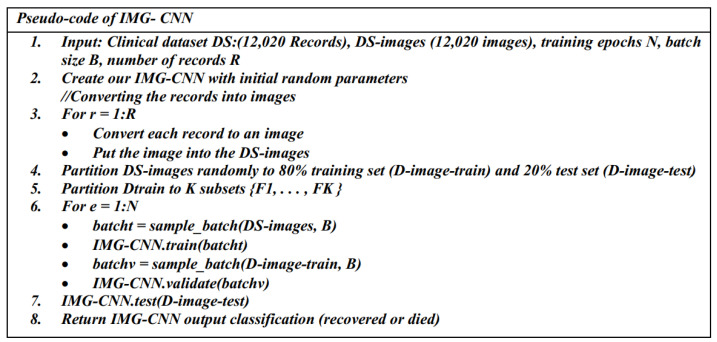
The pseudo-code of the IMG-CNN model.

**Figure 11 diagnostics-12-01847-f011:**
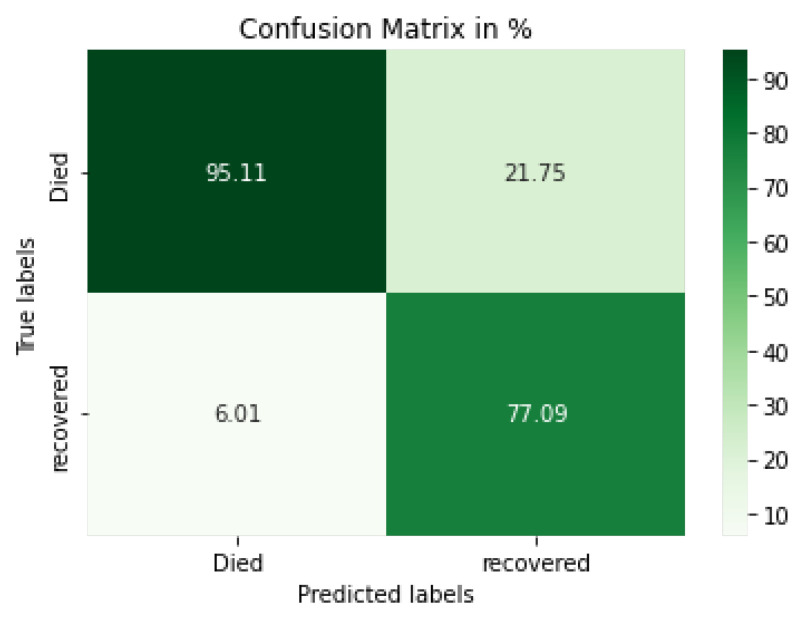
Confusion matrix for the proposed CV-CNN model.

**Figure 12 diagnostics-12-01847-f012:**
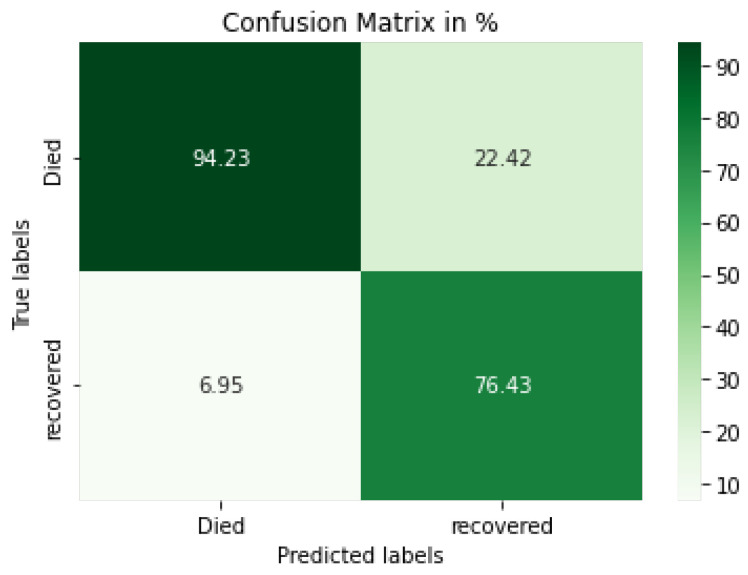
Confusion matrix for the proposed LSTM+CNN model.

**Figure 13 diagnostics-12-01847-f013:**
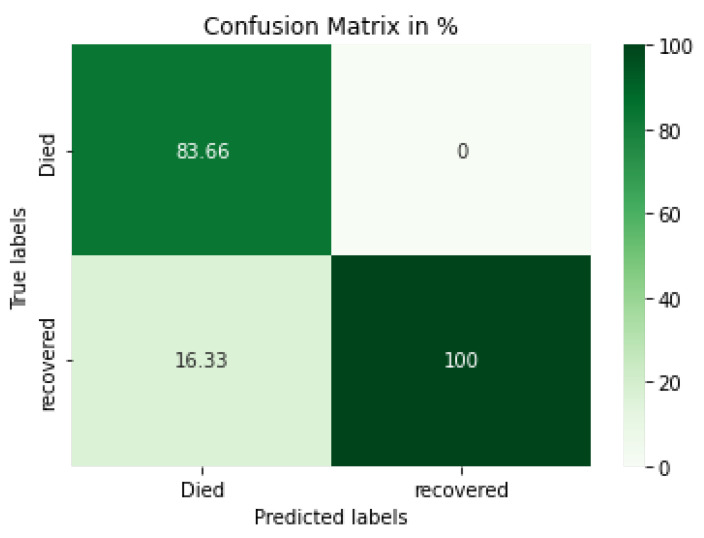
Confusion matrix for the proposed IMG-CNN model.

**Figure 14 diagnostics-12-01847-f014:**
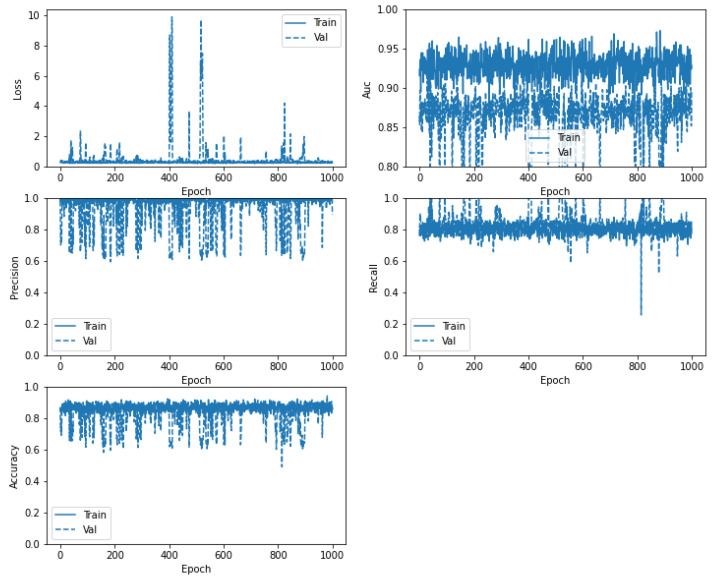
Loss, AUC, precision, recall, and accuracy between the training and validation phases, with the number of epochs for the IMG-CNN model.

**Figure 15 diagnostics-12-01847-f015:**
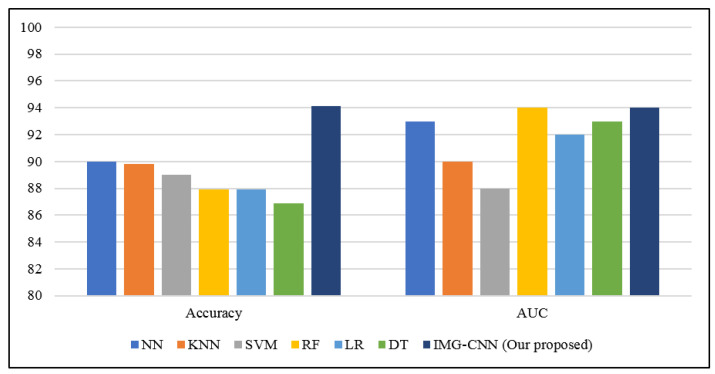
Comparison of the performance evaluation metrics of the different models presented in previous studies (NN, KNN, SVM, RF, LR, and DT) and our proposed IMG-CNN model using the same clinical dataset.

**Figure 16 diagnostics-12-01847-f016:**
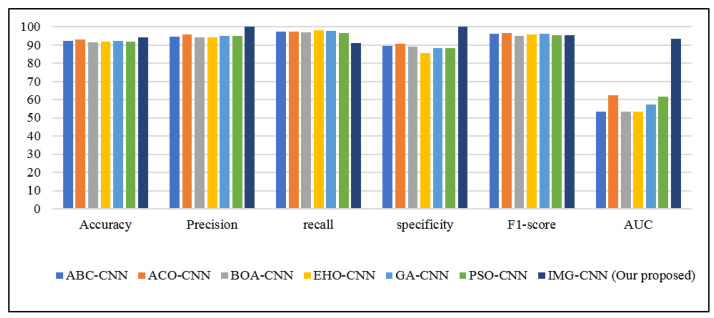
Comparison of the performance evaluation metrics of the different models presented in previous studies and our proposed IMG-CNN model using deep learning models.

**Figure 17 diagnostics-12-01847-f017:**
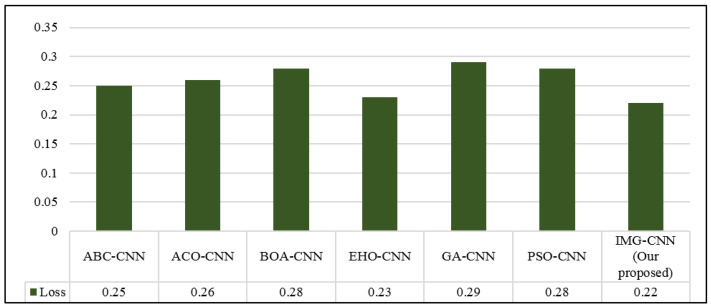
Loss comparison among the different models presented in previous studies and our proposed IMG-CNN model using deep learning models.

**Table 1 diagnostics-12-01847-t001:** Summary of the commonly reviewed studies on the severity and the prediction of mortality risk in COVID-19 patients.

Study	Method	ML/DL	Performance
Pourhomayoun et al. [1]	SVM, NN, and RF	Machine learning	89.98% (Accuracy)
Yan et al. [19]	XGBoost	Machine learning	Accuracy 90%
Yan et al. [27]	XGBoost	Machine learning	93% (Accuracy)
Wang et al. [28]	XGBoost	Machine learning	83% (AUC for clinical model) 88% (AUC for laboratory model)
Rechtman et al. [29]	XGBoost	Machine learning	86% (AUC)
Bertsimas et al. [30]	XGBoost	Machine learning	81%, 87%, and 92% (AUCs using three validation cohorts)
Guan et al. [31]	XGBoost	Machine learning	Precision >90%, Sensitivity >85% F1 scores 0.9
Booth et al. [32]	SVM	Machine learning	93% (AUC), 91% (Specificity), 91 % (Sensitivity)
Sun et al. [33]	SVM	Machine learning	97.57% (AUC)
Yao et al. [34]	SVM	Machine learning	81.48% (Accuracy)
Zhao et al. [35]	SVM	Machine learning	91.38% (Accuracy), 94% (Specificity) 90 % (Sensitivity)
Chowdhury et al. [20]	XGBoost	Machine learning	96.1% (AUC)
Karthikeyan et al. [21]	NN, SVM, LR, random forests, XGBoost, and DT	Machine learning	NN model performance 96.53% (Accuracy), 98.9% (AUC), 96.9 % (F1 scores)
Kar et al. [22]	XGBoost	Machine learning	78.2% (AUC), 93% (Accuracy score), 77% (Precision)
Hu et al. [36]	partial least squares regression, elastic net model, RF, bagged FDA, and LR	Machine learning	LR model performance 88.1% (AUC), 79.4% (Specificity), 83.9% (Sensitivity)
Zhao et al. [37]	LR	Machine learning	83% (AUC for mortality prediction), 74% (AUC for ICU admission prediction)
Huang et al. [38]	LR	Machine learning	94.4% (AUC), 90.2% (Specificity), 94.1% (Sensitivity)
Zhou et al. [39]	LR	Machine learning	87.9% (AUC), 73.7% (Specificity), 88.6% (Sensitivity)
Zhu et al. [40]	LR	Machine learning	90% (AUC)
Gong et al. [41]	LASSO regression, DT, RF, SVM, and LR	Machine learning	LR model performance 85.3% (AUC), 78.4% (Specificity) 77.5% (Sensitivity)
Liu et al [43]	Multivariable LR	Machine learning	99.4% (AUC), 97.2% (Specificity) 100% (Sensitivity)
Li et al. [45]	Autoencoder, LR, RF, SVM, one-class SVM, isolation forest, and local outlier factor	Machine learning	Autoencoder model performance 97% (Accuracy) and 73% (AUC)
Terwangne et al. [46]	Bayesian network analysis	Machine learning	83.8% (ROC for WHO classification model) and 91% (ROC for EPI-SCORE model),
Aktar et al. [23]	Random Forest, DT, GBM, XGBoost, SVM, and LGBM	Machine learning	88% (Accuracy of comorbidity and mortality for LGBM model), 90% (Accuracy of symptoms for GBM and LGBM models)
Tezza et al. [24]	RPART, SVM, GBM, and Random Forest	Machine learning	Random Forest model performance 84% (ROC)
Khozeimeh et al. [25]	CNN and autoencoders	Deep learning	96.05% (Average Accuracy)

**Table 2 diagnostics-12-01847-t002:** The list of the 57 selected features used in our predictive models.

Feature Type	Feature Name		
Symptoms	anorexia	fever	shortness of breath
	chest pain	gasp	somnolence
	chills	headache	sore throat
	conjunctivitis	kidney failure	sputum
	cough	lesions on chest radiographs	septic shock
	diarrhea	hypertension	Heart attack
	dizziness	Myalgia	old
	dyspnea	Obnubilation	cardiac disease
	emesis	pneumonia	hypoxia
	expectoration	myelofibrosis	fatigue
	eye irritation	respiratory distress	rhinorrhea
Pre-existing Conditions	diabetes	COPD	coronary heart disease
	hypertension	Parkinson’s disease	prostate hypertrophy
	chronic kidney disease	asthma	Tuberculosis
	hypothyroidism	cancer	hepatitis B
	cerebral infarction	HIV positive	chronic bronchitis
	cardiac disease	dyslipidemia	any chronic disease
Demographics	age	country	province
	gender	city	travel history

**Table 3 diagnostics-12-01847-t003:** Architecture and parameter settings of the CV-CNN model.

Layer (Type)	Output Shape	Parameters
conv1d_5 (Conv1D)	(None, 52, 256)	1024
conv1d_6 (Conv1D)	(None, 50, 256)	196,864
conv1d_7 (Conv1D)	(None, 48, 256)	196,864
max_pooling1d_1 (MaxPooling1D)	(None, 47, 256)	0
flatten_2 (Flatten)	(None, 12,032)	0
dense_6 (Dense)	(None, 64)	770,112
batch_normalization_4 (BatchNormalization)	(None, 64)	256
dropout_4 (Dropout)	(None, 64)	0
dense_7 (Dense)	(None, 32)	2080
batch_normalization_5 (BatchNormalization)	(None, 32)	128
dropout_5 (Dropout)	(None, 32)	0
dense_8 (Dense)	(None, 1)	33
Total parameters: 1,167,361		
Trainable parameters: 1,167,169		
Non-trainable parameters: 192		

**Table 4 diagnostics-12-01847-t004:** Architecture and parameter settings of the CV-LSTM+CNN model.

Layer (Type)	Output Shape	Parameters
batch_normalization_2 (BatchNormalization)	(None, 54, 1)	4
Reshape (Reshape)	(None, 9, 6, 1)	0
Time_distribution (TimeDistributed)	(None, 9, 6, 256)	264,192
dropout_2 (Dropout)	(None, 9, 6, 256)	0
batch_normalization_3 (BatchNormalization)	(None, 9, 6, 256)	1024
Time_distribution_1 (TimeDistributed)	(None, 9, 6, 256)	26,2400
conv1d_4 (Conv1D)	(None, 9, 4, 256)	196,864
average_pooling2d_1 (AveragePooling2D)	(None, 4, 2, 256)	0
flatten_1 (Flatten)	(None, 2048)	0
dropout_3 (Dropout)	(None, 2048)	0
dense_3 (Dense)	(None, 128)	262,272
dense_4 (Dense)	(None, 64)	8256
dense_5 (Dense)	(None, 1)	65
Total parameters: 995,077		
Trainable parameters: 994,563		
Non-trainable parameters: 514		

**Table 5 diagnostics-12-01847-t005:** Architecture and parameter settings of the IMG-CNN model.

Layer (Type)	Output Shape	Parameters
Conv2d (Conv2D)	(None, 224, 224, 256)	7168
Activation (Activation)	(None, 224, 224, 256)	0
batch_normalization (BatchNormalization)	(None, 224, 224, 256)	1024
Conv2d_1 (Conv2D)	(None, 224, 224, 128)	295,040
Activation_1 (Activation)	(None, 224, 224, 128)	0
max_pooling2d (MaxPooling2D)	(None, 74, 74, 128)	0
dropout (Dropout)	(None, 74, 74, 128)	0
Conv2d_2 (Conv2D)	(None, 72, 72, 64)	73,792
Activation_2 (Activation)	(None, 72, 72, 64)	0
batch_normalization_1 (BatchNormalization)	(None, 72, 72, 64)	256
flatten (Flatten)	(None, 331,776)	0
dense (Dense)	(None, 512)	169,869,824
dropout_1 (Dropout)	(None, 512)	0
dense_1 (Dense)	(None, 64)	32,832
dense_2 (Dense)	(None, 1)	65
Total parameters: 170,280,001		
Trainable parameters: 170,279,361		
Non-trainable parameters: 640		

**Table 6 diagnostics-12-01847-t006:** Results of the CV-CNN proposed model using different evaluation metrics based on a 10-fold cross-validation.

Fold	Precision Recovered	Died	Macro	Weight	Recall Recovered	Died	Macro	Weight	F1-Score Recovered	Died	Macro	Weight	Accuracy
1	0.79	0.93	0.86	0.86	0.94	0.75	0.85	0.85	0.86	0.83	0.85	0.85	0.85
2	0.80	0.93	0.86	0.86	0.94	0.76	0.85	0.85	0.86	0.83	0.85	0.85	0.85
3	0.53	0.95	0.74	0.74	1.00	0.09	0.54	0.55	0.69	0.17	0.43	0.43	0.55
4	0.59	0.95	0.77	0.77	0.98	0.30	0.64	0.64	0.74	0.46	0.60	0.60	0.64
5	0.69	0.93	0.81	0.81	0.96	0.57	0.77	0.77	0.80	0.71	0.76	0.76	0.77
6	0.74	0.93	0.84	0.83	0.95	0.66	0.80	0.81	0.83	0.77	0.80	0.80	0.81
7	0.81	0.93	0.87	0.87	0.94	0.78	0.86	0.86	0.87	0.85	0.86	0.86	0.86
8	0.50	0.00	0.25	0.25	1.00	0.00	0.50	0.50	0.67	0.00	0.33	0.34	0.50
9	0.81	0.92	0.87	0.87	0.94	0.77	0.86	0.86	0.87	0.84	0.86	0.86	0.86
10	0.65	0.92	0.78	0.78	0.96	0.47	0.71	0.72	0.77	0.62	070	0.70	0.72

**Table 7 diagnostics-12-01847-t007:** Results of the LSTM + CNN proposed model using different evaluation metrics based on a 10-fold cross-validation.

Fold	Precision Recovered	Died	Macro	Weight	Recall Recovered	Died	Macro	Weight	F1-Score Recovered	Died	Macro	Weight	Accuracy
1	0.81	0.93	0.87	0.87	0.94	0.77	0.86	0.86	0.87	0.84	0.85	0.85	0.86
2	0.83	0.63	0.73	0.73	0.48	0.90	0.69	0.69	0.61	0.74	0.68	0.67	0.69
3	0.81	0.92	0.86	0.86	0.93	0.78	0.85	0.85	0.87	0.84	0.85	0.85	0.85
4	0.76	0.61	0.68	0.69	0.47	0.85	0.66	0.66	0.58	0.71	0.65	0.65	0.66
5	0.81	0.93	0.87	0.87	0.94	0.77	0.86	0.86	0.87	0.84	0.86	0.86	0.86
6	0.81	0.92	0.86	0.86	0.93	0.77	0.85	0.85	0.86	0.84	0.85	0.85	0.85
7	0.63	0.50	0.56	0.56	0.01	0.99	0.50	0.50	0.03	0.66	0.35	0.34	0.50
8	0.96	0.50	0.73	0.73	0.02	1.00	0.51	0.51	0.04	0.67	0.35	0.35	0.51
9	0.79	0.92	0.86	0.86	0.94	0.75	0.84	0.84	0.86	0.83	0.84	0.84	0.84
10	0.10	0.49	0.30	0.30	0.00	0.98	0.49	0.49	0.00	0.66	0.33	0.33	0.49

**Table 8 diagnostics-12-01847-t008:** Evaluation metrics for the IMG-CNN model.

Performance Metric	Value	Performance Metric	Value	Performance Metric	Value	Performance Metric	Value
Tp	124	val_Tp	305	accuracy	0.85	val_accuracy	0.94
Fp	2	val_Fp	0	precision	0.98	val_precision	1
Tn	94	val_Tn	177	recall	0.77	val_recall	0.91
Fn	36	val_Fn	30	AUC	0.94	val_AUC	0.936
loss	0.27	val_loss	0.22				

**Table 9 diagnostics-12-01847-t009:** Evaluation metrics for the three proposed Deep-Risk models.

Models	Precision Recovered	Died	Avg.	Recall Recovered	Died	Avg.	F1-Score Recovered	Died	Avg.	Accuracy
CV-LSTM + CNN	81%	92%	86.5%	93%	77%	85%	86%	84%	85%	85.27%
CV-CNN	81%	93%	87%	94%	78%	86%	87%	85%	86%	86.06%
IMG-CNN	83%	100%	91.5%	100%	84%	92%	91%	91%	91%	94.14%

**Table 10 diagnostics-12-01847-t010:** Comparison of the performance evaluation metrics of the different models presented in previous studies and our proposed IMG-CNN model using the same clinical dataset.

Models	Rank	Accuracy (%)	AUC (%)
NN [1]	2	89.98	93
KNN [1]	3	89.83	90
SVM [1]	4	89.02	88
RF [1]	5	87.93	94
LR [1]	6	87.91	92
DT [1]	7	86.87	93
IMG-CNN (proposed)	1	94.14	93.6

**Table 11 diagnostics-12-01847-t011:** Comparison of the performance evaluation metrics of the different models presented in previous studies and our proposed IMG-CNN model using deep learning models.

Models	Rank	Accuracy (%)	Precision (%)	Recall (%)	Specificity (%)	F1-score (%)	AUC (%)	Loss
ABC-CNN [25]	3	92.32	94.7	97.4	89.65	96.0	53.3	0.25
ACO-CNN [25]	2	93.10	95.6	97.3	90.57	96.4	62.5	0.26
BOA-CNN [25]	7	91.37	94.1	97.0	89.08	95.1	53.5	0.28
EHO-CNN [25]	5	91.86	94.1	98.0	85.69	95.9	53.2	0.23
GA-CNN [25]	4	92.18	94.8	97.8	88.32	96.1	57.5	0.29
PSO-CNN [25]	6	91.85	95.0	96.4	88.17	95.5	61.5	0.28
IMG-CNN (proposed)	1	94.14	100	91.0	100	95.3	93.6	0.22

## Data Availability

Not applicable.

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
