# Peer review of "Deep-Risk: Deep Learning-Based Mortality Risk Predictive Models for COVID-19"

_diagnostics, 2022, doi:10.3390/diagnostics12081847_

Round 1
Reviewer 1 Report
The authors supervised deep learning system for COVID-19 risk mortality prediction is developed, called the Deep-Risk system. Three deep learning models were designed to solve this problem: CV-CNN model, CV-LSTM+CNN model, and IMG-CNN model. Each model is trained and tested to see how well they can be able to identify the cases that have been recovered and those that have died
This is a very nice manuscript. I have only a few points which deserve clarification.
The introduction section is too long. I suggest to short it
The discussion is feeble. I recommend amplifying it. The author should discuss the other method, which gives some information about the prognosis (please cite the following paper PubMed ID 33512742)
In the discussion, the authors can also discuss treatment (please cite the following paper DOI 10.3389/fphar.2020.01124)
Author Response
Journal: Diagnostics
Manuscript ID: diagnostics-1788845
Title: ' Deep-Risk: Deep learning-based Mortality Risk Predictive models for COVID-19’
Response letter for Reviewer 1
We would like to thank the academic reviewers for the valuable feedback and his precise very constructive
comments. Also, we would like to thank the reviewers for the smooth coordination of the review process. We
are happy to discuss further if any issues still are subject to further questions.
As requested, we have responded to all the comments in detail and suggested improvements for the revised
manuscript.
In the following, we responded to the comments of reviewer 1
Reviewer 1:
Thank you for your helpful comments and for taking the time to comment on my paper. It improved my
work remarkably.
The authors supervised deep learning system for COVID-19 risk mortality prediction is developed,
called the Deep-Risk system. Three deep learning models were designed to solve this problem: CVCNN model, CV-LSTM+CNN model, and IMG-CNN model. Each model is trained and tested to see
how well they can be able to identify the cases that have been recovered and those that have died
This is a very nice manuscript. I have only a few points which deserve clarification.
1- The introduction section is too long. I suggest to short it
The introduction section has been shortening in the manuscript
2- The discussion is feeble. I recommend amplifying it.
4.3. Deep-Risk: deep learning-based risk mortality prediction models results
Our response is to add the following text:
As we introduce three proposed deep learning predictive models for risk in Covid-19 patients, we begin
our results by examining the performance of the three models in terms of precision, recall, F1-score
and accuracy using the dataset given in [1]. A complete analysis of these models is given indicating
their performance in predicting recovered and died patients. The confusion matrix is also given for each
model and the behaviour of the model during training through epochs are presented as graphs for
precision, recall, loss, AUC and accuracy. The results of the proposed Deep-Risk models are presented
followed by a brief discussion and analysis for each proposed model.
4.4. Experimental comparisons
Replace by
4.4. Proposed Models Comparison and Results Discussion
The current research focused on the use of deep learning systems to predict the survival chances of
COVID-19 patients. We used patient data consisting of 12,020 records from the clinical dataset to
construct and evaluate the proposed models [1]. First, this clinical dataset in its original numerical form
was used to train and test two of the proposed models (CV-CNN and CV-LSTM+CNN). Then, secondly,
all these records were converted into images so that each image represents one row of data from the
original dataset and was also used to train and test the third proposed model (IMG-CNN model).
The experimental results of the three proposed Deep-Risk models are summarized in Table 9. The
results of the CV-CNN and the CV-LSTM+CNN models for each performance metric were computed
as the average of all the 10 folds. As clearly seen from Table 9, IMG-CNN model is much better than
the other two proposed models in terms of accuracy, precision, F1-score, and accuracy. The use of the
image form in the clinical dataset gives better results than using the original dataset especially with
deep learning models.
Examining the results, we found that there were significant differences in performance between CNNs
trained on clinical data and those trained on image data. The reason is that the deep learning approach
can identify the features presented in the input data more precise when the input is an image, as the
operations performed on the input works better with images. This explains why converting the data into
images and using it in the IMG-CNN model yields better results. Accordingly, the proposed IMG-CNN
model will be used in the next section and compared with the previous models presented in previous
work.
5. Comparative Analysis with State-of the-art Work
In our comparative analysis, the previous studies concerned with predicting the survival chances of
COVID-19 patients were divided into two aspects: first, comparison with previous work that used the
same dataset we used in our experiments but their models were based machine learning methods.
Second, comparison with previous work that used deep learning-based (CNN model) but with different
descriptive methods and dataset. Besides, we also present the analysis of the experimental results to
evaluate the prediction ability of our model.
5.1. Dataset-based Comparison with State-of the-art Work
As discussed in the literature review section, very few methods have focused on predicting mortality
using clinical data. Additionally, existing methods have used different features than the used in our
experiments. In this section, we compare our work against the recent work that used ML approaches
namely; Support Vector Machine, Neural Network, KNN, LR, DT, and Random Forest on the same
dataset [1]. As concluded from the results presented in Table 9, in the previous section, the proposed
IMG-CNN model gives better performance that the other proposed models, therefore, we will use only
the proposed IMG-CNN model in the comparison with the previous study that used the same clinical
dataset [1].
Table 10 demonstrates the performance evaluation metrics comparison among the different models
presented in the previous study [1] (as reported in their paper) and our proposed IMG-CNN model using
the same clinical dataset, as also represented in Figure 15. In the present paper, we evaluated the
proposed model in terms of accuracy and AUC. The accuracy reached by our IMG-CNN model is
94.14% and the AUC is 93.6% compared to the best results reported in [1] of the NN model which were
89.98% and 93% in accuracy and AUC, respectively. The results show that our proposed IMG-CNN
model outperforms all the existing models by a significant value which demonstrates the effectiveness
of our predictive model. This is due to the fact that deep learning-based models are more accurate than
the machine learning-based models as they are able to extract the important features of the input data
in an accurate automatic manner.
5.2. Deep learning-based Comparison with State-of the-art Work
There are few methods that have studied mortality rate prediction based on clinical data using deep
learning models. However, the features and the clinical dataset which used in our experiments differ
from those used in the introduced study [25].
Because the proposed IMG-CNN model outperforms the other two proposed models, as shown in Table
9, it will be used in the comparison of the previous studies that used deep learning models. Table 11
shows the performance evaluation metrics of the various models presented in previous study [25] (as
reported by their paper) and our proposed IMG-CNN model in terms of accuracy, precision, recall,
specificity, F1-score, AUC, and loss. Figure 16 and Figure 17 show their results in a graphical form for
a clearer visual comparison. The conducted experiments results reveal that our proposed IMG-CNN
model outperforms the previous studies with a significant percentage in accuracy, precision, specificity,
and AUC. It achieved the highest accuracy (94.14%), precision (100%), specificity (100%), and AUC
(93.6%), and the minimum loss. This is due the better representation of the features in the image form
which allowed the deep learning model to identify the correlation between the input features of the data
producing better prediction ability.
3- The author should discuss the other method, which gives some information about the prognosis
(please cite the following paper PubMed ID 33512742)
The suggested reference is:
Bergamaschi, L., D’Angelo, E.C., Paolisso, P., Toniolo, S., Fabrizio, M., Angeli, F., Donati, F., Magnani,
I., Rinaldi, A., Bartoli, L. and Chiti, C., 2021. The value of ECG changes in risk stratification of COVID‐
19 patients. Annals of Noninvasive Electrocardiology, 26(3), p.e12815., doi: 10.1111/anec.12815
We added the following text
Several medical studies, such as [ref requested], have exploited the prognosis in SARS-CoV-2
hospitalized patients. They evaluated the ECG readings, as an indicator of heart disease, at admission
to the hospital and after 7 days of hospitalization in SARS-CoV-2 hospitalized patients and concluded
that ECG is useful in identifying patients with possible clinical risk. There was also a significant
association between abnormal ECG and major adverse events in patients with COVID-19.
4- In the discussion, the authors can also discuss treatment (please cite the following paper DOI
10.3389/fphar.2020.01124)
The suggested reference is:
Paolisso, P., Bergamaschi, L., D’Angelo, E.C., Donati, F., Giannella, M., Tedeschi, S., Pascale, R.,
Bartoletti, M., Tesini, G., Biffi, M. and Cosmi, B., 2020. Preliminary experience with low molecular weight
heparin strategy in COVID-19 patients. Frontiers in pharmacology, 11, p.1124. doi:
10.3389/fphar.2020.01124
We added the following text
On the other hand, doctors have been seeking the best possible medicine dosages that will result in a
fast and effective cure for the patients. Studies have been presented to foresee the effect of certain
dosages in reducing the mortality risk as in [ref requested] which investigated the possible association
between different dosages of LMWH enoxaparin administration and mortality in hospitalized COVID-19
patients.
The manuscript English grammar, spelling, punctuation, and some improvement of
style were edited and checked
We hope that these answers will be forwarded to the academic reviewer and answer most of his questions.
We are happy to discuss further if any issues still are subject to further questions
Thank you very much for your efforts,
Nada Elshennawy on behalf of all authors

Reviewer 2 Report
The manuscript "Deep-Risk: Deep learning-based Mortality Risk Predictive models for COVID-19" is an interesting contribution showing novelty and nice methods. Nevertheless it requires improvement in many places to eliminate misunderstandings and the presentation of information.
please rewrite the abstract, some sentences seem not to be appropriate, like "Despite it has low mortality rate" - this claim is disputable.
W would suggest to rewrite and rearrange the Figure 2, instead of showing a snapshot of the original clinical dataset.
line 399. The authors write that among 112 features, only 57 are selected. What what the criteria for this selection.
Figure 4 is not informative. It should be replaced or better discussed and presented.
line 581 and 582. Please rewrite and better explain
many editing mistakes and grammatical mistakes, some examples in lines 36-38, 500.
the process of converting the original clinical dataset into images should be better explained and discussed.
line 359 and 360, what do the authors mean? should it be machine learning vs deep learning or shallow methods vs deep learning? Please rewrite.
Please rewrite line 363 and 364 "the original and image forms of the clinical dataset can be used to feed the CNN models and improve their performance". Have these both datasets been used together to make the training dataset bigger in order to improve the performance? If so this is the same dataset but used twice. Please better explain. Also please explain the advantages of image form of the dataset over the original form.
what do the authors mean in line 366 "to fill this gap in knowledge"?
line 567 - when you refer to the virus, please write SARS-CoV-2 virus instead of COVID-virus, because this expression is not appropriate.
The Discussion section mainly focuses on the comparison the the models performance but lacks a simple summary of the medical data that is crucial, for the assessment of the patient's outcome. Please focus on the medical aspects so that the medical audience would also benefit from the lecture.
Also what do you mean by "previously unexplored aspects of the COVID-virus" in line 567? If some aspects of the disease have been uncovered please describe it.
Author Response
Journal: Diagnostics
Manuscript ID: diagnostics-1788845
Title: ' Deep-Risk: Deep learning-based Mortality Risk Predictive models for COVID-19’
Response letter for Reviewer 2
We would like to thank the academic reviewers for the valuable feedback and his precise very constructive
comments. Also, we would like to thank the reviewers for the smooth coordination of the review process. We
are happy to discuss further if any issues still are subject to further questions.
As requested, we have responded to all the comments in detail and suggested improvements for the revised
manuscript.
In the following, we responded to the comments of reviewer 2
Reviewer 2
Thank you for your helpful comments and for taking the time to comment on my paper. It improved my
work remarkably.
The manuscript "Deep-Risk: Deep learning-based Mortality Risk Predictive models for COVID-19" is
an interesting contribution showing novelty and nice methods. Nevertheless it requires improvement
in many places to eliminate misunderstandings and the presentation of information.
| 1- | please rewrite the abstract, some sentences seem not to be appropriate, like "Despite it has low mortality rate" - this claim is disputable. |
The abstract has been rewritten
2- We would suggest to rewrite and rearrange the Figure 2, instead of showing a snapshot of
the original clinical dataset.
The figure was rewritten and rearranged as follows:
| Age | Sex | Anorexia | Cough | Fatigue | Heart Attack | Septic Shock | cold | Outcome |
| 43 | 1 | 0 | 1 | 1 | 0 | 0 | 0 | Died |
| 65 | 0 | 0 | 0 | 0 | 0 | 0 | 0 | Recovered |
| 43 | 1 | 0 | 1 | 0 | 0 | 0 | 1 | Recovered |
Figure 2: Sample of the tabular dataset
| 3- | line 399. The authors write that among 112 features, only 57 are selected. What what the criteria for this selection. |
The used dataset was processed in the research paper [1] by filter and wrapper methods to rank
the best feature subsets. [1] concluded that these 57 from 112 were the most suitable features to
be used in COVID-19 mortality prediction.
4- Figure 4 is not informative. It should be replaced or better discussed and presented.
In the paper: These converted images will be used as inputs for the IMG-CNN model. Samples of
image representation of the dataset are shown in Figure 4
Replace with: When the tabular data is converted to images, the numbers are converted to pixel
colors in the image, ranging from black to white. The image appears with a black part and a white
part based on the values of the numbers in the data. These converted images will be used as
inputs for the IMG-CNN model. Samples of converted images are shown in Figure 4
| 5- | line 581 and 582. Please rewrite and better explain Done |
6- many editing mistakes and grammatical mistakes, some examples in lines 36-38, 500.
Done
7- the process of converting the original clinical dataset into images should be better explained
and discussed.
The tabular data to image conversion was made by some python and CV2 functions. Each sample in
the tabular data was converted into an image which has pixels and pixel intensities according to the
feature and its value, respectively. The same feature is represented by the same pixel (or pixels) in the
image for all original data samples with varying in the pixel intensities according to the feature value.
| 8- | line 359 and 360, what do the authors mean? should it be machine learning vs deep learning or shallow methods vs deep learning? Please rewrite. |
Please rewrite line 363 and 364 "the original and image forms of the clinical dataset can be
used to feed the CNN models and improve their performance". Have these both datasets
been used together to make the training dataset bigger in order to improve the performance?
If so this is the same dataset but used twice. Please better explain. Also please explain the
advantages of image form of the dataset over the original form. Also please explain the
advantages of image form of the dataset over the original form.
Our response is to add the following text:
The dataset that was gathered is represented in two ways. The first version is the original form, which
contains 12,020 records of patients who tested positive for Covid-19. The second form is created by
turning every record from the first form into images, with each image standing in for a row of data in the
initial clinical dataset.
The original data (Tabular data) was used as it is in CNN model and these data was converted to
images then used as inputs for 2-dimentional CNN in IMG-CNN model.
The main reason for converting the clinical dataset to images is to use the 2-dimensional convolutional
neural networks rather than 1-dimension convolutional neural networks. 2-dimensional convolutional
neural networks have many advantages: extracting the spatial features from data and making robust
network for classification. Also tabular data do not have a spatial relationship between features, tabular
data was preferred to converted to images to be more stable with the CNN architecture.
9- what do the authors mean in line 366 "to fill this gap in knowledge"?
Done
10- line 567 - when you refer to the virus, please write SARS-CoV-2 virus instead of COVID-virus,
because this expression is not appropriate.
Done
11- The Discussion section mainly focuses on the comparison the models performance but lacks
a simple summary of the medical data that is crucial, for the assessment of the patient's
outcome. Please focus on the medical aspects so that the medical audience would also
benefit from the lecture.
The Discussion section was rewritten and replaced by the following:
4.3. Deep-Risk: deep learning-based risk mortality prediction model results
Our response is to add the following text:
As we introduce three proposed deep learning predictive models for risk in Covid-19 patients, we begin
our results by examining the performance of the three models in terms of precision, recall, F1-score
and accuracy using the dataset given in [1]. A complete analysis of these models is given indicating
their performance in predicting recovered and died patients. The confusion matrix is also given for each
model and the behaviour of the model during training through epochs are presented as graphs for
precision, recall, loss, AUC and accuracy. The results of the proposed Deep-Risk models are presented
followed by a brief discussion and analysis for each proposed model.
4.4. Experimental comparisons
Replace by
4.4. Proposed Models Comparison and Results Discussion
The current research focused on the use of deep learning systems to predict the survival chances of
COVID-19 patients. We used patient data consisting of 12,020 records from the clinical dataset to
construct and evaluate the proposed models [1]. First, this clinical dataset in its original numerical form
was used to train and test two of the proposed models (CV-CNN and CV-LSTM+CNN). Then, secondly,
all these records were converted into images so that each image represents one row of data from the
original dataset and was also used to train and test the third proposed model (IMG-CNN model).
The experimental results of the three proposed Deep-Risk models are summarized in Table 9. The
results of the CV-CNN and the CV-LSTM+CNN models for each performance metric were computed
as the average of all the 10 folds. As clearly seen from Table 9, IMG-CNN model is much better than
the other two proposed models in terms of accuracy, precision, F1-score, and accuracy. The use of the
image form in the clinical dataset gives better results than using the original dataset especially with
deep learning models.
Examining the results, we found that there were significant differences in performance between CNNs
trained on clinical data and those trained on image data. The reason is that the deep learning approach
can identify the features presented in the input data more precise when the input is an image, as the
operations performed on the input works better with images. This explains why converting the data into
images and using it in the IMG-CNN model yields better results. Accordingly, the proposed IMG-CNN
model will be used in the next section and compared with the previous models presented in previous
work.
5. Comparative Analysis with State-of the-art Work
In our comparative analysis, the previous studies concerned with predicting the survival chances of
COVID-19 patients were divided into two aspects: first, comparison with previous work that used the
same dataset we used in our experiments but their models were based machine learning methods.
Second, comparison with previous work that used deep learning-based (CNN model) but with different
descriptive methods and dataset. Besides, we also present the analysis of the experimental results to
evaluate the prediction ability of our model.
5.1. Dataset-based Comparison with State-of the-art Work
As discussed in the literature review section, very few methods have focused on predicting mortality
using clinical data. Additionally, existing methods have used different features than the used in our
experiments. In this section, we compare our work against the recent work that used ML approaches
namely; Support Vector Machine, Neural Network, KNN, LR, DT, and Random Forest on the same
dataset [1]. As concluded from the results presented in Table 9, in the previous section, the proposed
IMG-CNN model gives better performance that the other proposed models, therefore, we will use only
the proposed IMG-CNN model in the comparison with the previous study that used the same clinical
dataset [1].
Table 10 demonstrates the performance evaluation metrics comparison among the different models
presented in the previous study [1] (as reported in their paper) and our proposed IMG-CNN model using
the same clinical dataset, as also represented in Figure 15. In the present paper, we evaluated the
proposed model in terms of accuracy and AUC. The accuracy reached by our IMG-CNN model is
94.14% and the AUC is 93.6% compared to the best results reported in [1] of the NN model which were
89.98% and 93% in accuracy and AUC, respectively. The results show that our proposed IMG-CNN
model outperforms all the existing models by a significant value which demonstrates the effectiveness
of our predictive model. This is due to the fact that deep learning-based models are more accurate than
the machine learning-based models as they are able to extract the important features of the input data
in an accurate automatic manner.
5.2. Deep learning-based Comparison with State-of the-art Work
There are few methods that have studied mortality rate prediction based on clinical data using deep
learning models. However, the features and the clinical dataset which used in our experiments differ
from those used in the introduced study [25].
Because the proposed IMG-CNN model outperforms the other two proposed models, as shown in Table
9, it will be used in the comparison of the previous studies that used deep learning models. Table 11
shows the performance evaluation metrics of the various models presented in previous study [25] (as
reported by their paper) and our proposed IMG-CNN model in terms of accuracy, precision, recall,
specificity, F1-score, AUC, and loss. Figure 16 and Figure 17 show their results in a graphical form for
a clearer visual comparison. The conducted experiments results reveal that our proposed IMG-CNN
model outperforms the previous studies with a significant percentage in accuracy, precision, specificity,
and AUC. It achieved the highest accuracy (94.14%), precision (100%), specificity (100%), and AUC
(93.6%), and the minimum loss. This is due the better representation of the features in the image form
which allowed the deep learning model to identify the correlation between the input features of the data
producing better prediction ability.
12- Also what do you mean by "previously unexplored aspects of the COVID-virus" in line 567? If
some aspects of the disease have been uncovered please describe it.
This statement has been modified in the manuscript.
The manuscript English grammar, spelling, punctuation, and some improvement of
style were edited and checked
We hope that these answers will be forwarded to the academic reviewer and answer most of his questions.
We are happy to discuss further if any issues still are subject to further questions
Thank you very much for your efforts,
Nada Elshennawy on behalf of all authors

Round 2
Reviewer 2 Report
I accept the current version of the manuscript for publication.